# Alternatives for MRI in Prostate Cancer Diagnostics—Review of Current Ultrasound-Based Techniques

**DOI:** 10.3390/cancers14081859

**Published:** 2022-04-07

**Authors:** Adam Gurwin, Kamil Kowalczyk, Klaudia Knecht-Gurwin, Paweł Stelmach, Łukasz Nowak, Wojciech Krajewski, Tomasz Szydełko, Bartosz Małkiewicz

**Affiliations:** 1University Center of Excellence in Urology, Department of Minimally Invasive and Robotic Urology, Wroclaw Medical University, 50-556 Wroclaw, Poland; kamil.kowalczyk@student.umw.edu.pl (K.K.); pawel.stelmach@umw.edu.pl (P.S.); lukasz.nowak@student.umw.edu.pl (Ł.N.); wojciech.krajewski@umw.edu.pl (W.K.); tomasz.szydelko@umw.edu.pl (T.S.); 2Department of Dermatology, Venereology and Allergology, Wroclaw Medical University, 50-368 Wroclaw, Poland; klaudia.knecht@student.umw.edu.pl

**Keywords:** prostate cancer, ultrasonography, biopsy, micro-ultrasound, elastography, contrast-enhanced ultrasound

## Abstract

**Simple Summary:**

Prostate cancer (PCa) is the most common solid malignant tumor in men worldwide with various clinical manifestations. Due to overdiagnosis and overtreatment of a clinically insignificant disease, multiparametric magnetic resonance imaging is recommended for every patient before performing prostate biopsy. However, the diagnostic pathway currently has many limitations and is still far from ideal. Therefore, further alternatives need to be investigated. As the novel ultrasound-based techniques, such as shear wave elastography, contrast-enhanced ultrasound or high frequency micro-ultrasound are able to, overcome the limitations of magnetic resonance imaging presenting good performance in recent studies, we have summarized and compared the results of each technique in the detection of PCa. Furthermore, we analyzed the future perspectives for ultrasound modalities that may soon significantly improve their diagnostic value.

**Abstract:**

The purpose of this review is to present the current role of ultrasound-based techniques in the diagnostic pathway of prostate cancer (PCa). With overdiagnosis and overtreatment of a clinically insignificant PCa over the past years, multiparametric magnetic resonance imaging (mpMRI) started to be recommended for every patient suspected of PCa before performing a biopsy. It enabled targeted sampling of the suspicious prostate regions, improving the accuracy of the traditional systematic biopsy. However, mpMRI is associated with high costs, relatively low availability, long and separate procedure, or exposure to the contrast agent. The novel ultrasound modalities, such as shear wave elastography (SWE), contrast-enhanced ultrasound (CEUS), or high frequency micro-ultrasound (MicroUS), may be capable of maintaining the performance of mpMRI without its limitations. Moreover, the real-time lesion visualization during biopsy would significantly simplify the diagnostic process. Another value of these new techniques is the ability to enhance the performance of mpMRI by creating the image fusion of multiple modalities. Such models might be further analyzed by artificial intelligence to mark the regions of interest for investigators and help to decide about the biopsy indications. The dynamic development and promising results of new ultrasound-based techniques should encourage researchers to thoroughly study their utilization in prostate imaging.

## 1. Introduction

Prostate cancer (PCa) is the most common male solid malignant tumor and the second leading cause of cancer-related death in men worldwide [1]. In PCa, various types of histopathologic and molecular heterogeneity have been observed by pathologists. Theoretically, the same disease differs clinically, ranging from indolent—insignificant cancer that needs active surveillance—to very aggressive ones with rapid metastases and fatal outcome [2,3]. This fact contributes to the lack of an ideal standardized diagnostic process. Until 2020, serum prostate specific antigen (PSA) level and an abnormal digital rectal examination (DRE) with the follow of 12-core transrectal ultrasonography (TRUS)-guided systematic prostate biopsy (SBx) used to be the main diagnostic pathway [4]. An isolated elevation in PSA level could be the only reason to undergo SBx. However, it led to overdiagnosis and overtreatment of clinically insignificant disease (40–65% of performed biopsies were redundant) [5,6,7]. Many patients without malignancy or with indolent PCa were exposed to unnecessary side-effects of the biopsy, such as bleeding, pain, dysuria, or infection [8,9]. Moreover, SBx is known to leave 30% of clinically significant PCa (csPCa) undiagnosed [10].

Addressing this issue prompted researchers to seek new ways to improve the current biopsy indications. Multiple studies suggested better outcomes with the use of pre-biopsy multiparametric magnetic resonance imaging (mpMRI) or new ultrasound techniques, such as shear wave elastography (SWE), contrast-enhanced ultrasound (CEUS), or high frequency micro-ultrasound (MicroUS) [7,11,12,13,14,15,16,17]. As a result, crucial changes in the 2020 European Association of Urology (EAU) guidelines were established, promoting mpMRI, firstly to being recommended for every patient with PCa suspicion (elevated PSA/abnormal DRE) before performing the biopsy, and secondly, as a tool to detect csPCa in the active surveillance (AS) [18]. Furthermore, mpMRI allowed to develop a prostate suspicious region of interest (ROI) model which can be superimposed on the ultrasound images at the time of biopsy (MRI-US fusion targeted biopsy—FBx). Fbx was included in the EAU guidelines as an addition to SBx [19]. Despite the changes, PCa diagnosis is still a matter of discussion among physicians and has been developing dynamically over the past few years.

In this review, we summarize the current medical knowledge and analyze multiple retrospective and prospective studies of the role of the new ultrasound techniques in the diagnostic pathway of PCa, with a brief revision of mpMRI utilization. We compare the individual effectiveness of the MicroUS, SWE, CEUS, and mpMRI in the qualification of the patients with PCa suspicion for further invasive diagnostics, and as a suggested targeted biopsy guidance procedure. Finally, we focus on the utilization of the aforementioned techniques performed together before the biopsy, and the future perspectives, such as the new contrast agents, three-dimensional (3D) models of the fusion of mpMRI with novel ultrasound modalities, and the potential to implement artificial intelligence (AI) for image interpretation.

## 2. Evidence Acquisition

For the purposes of this narrative review, we conducted a comprehensive English language literature research for original and review articles using the Medline database and grey literature through January 2022. We searched for the combination of following terms: prostate cancer, prostate biopsy, fusion prostate biopsy, systematic prostate biopsy, targeted prostate biopsy, multiparametric MRI, micro-ultrasound, shear-wave elastography, contrast-enhanced ultrasound, multiparametric ultrasound, and ultrasound prostate imaging. We found 711 related articles, and the final number of papers selected for this manuscript was 197. Studies with the highest level of evidence and relevance to the discussed topics (135) were selected with the consensus of the authors.

## 3. Multiparametric MRI in the Prostate Cancer Diagnostics

As the traditional SBx has shown to be inadequate in the accurate diagnosis of PCa, the incorporation of mpMRI has helped to address some of the shortcomings [20,21]. The detection of PCa using mpMRI has faced a significant challenge in the development of standardized reporting system, which would reduce the difference in experience between radiologists. For this reason, in 2012, the PI-RADS v1 system was introduced, which applied a set of precise criteria to determine specific scores of cancer suspicion [22]. PI-RADS v1 became the first standardization of prostate mpMRI reporting. This system included a 5-point scale from “1” for very low suspicion to “5” for very high suspicion and was characterized by 78% sensitivity and 79% specificity [23]. Nevertheless, due to its complex and poor time efficient scoring chart, in 2015, it was upgraded to the PI-RADS v2 [24]. The system addressed the limitations of its predecessor simplifying the calculation of final score and the assignment of detected lesions. Furthermore, a meta-analysis based on 21 studies (3857 men) of the PI-RADS systems established an improvement of sensitivity (v2: 95% vs. v1: 88%, *p* < 0.05), but no significant changes in specificity (v2: 73% vs. v1: 75%, *p* = 0.90) for the PI-RADS v2 [25]. The meta-analysis demonstrated a cut-off suspicion score of ≥4 to provide acceptable sensitivity (89%) and specificity (74%). Decreasing the cut-off to ≥3 improved sensitivity (95%), but specificity decreased significantly (47%).

After the implementation of the PI-RADS system to the standard diagnostic routine, multiple studies about the utilization of mpMRI in PCa diagnostics were published. The comparison of mpMRI biopsy qualification and SBx was published in the multicenter prospective PROMIS trial [7]. This trial included 576 men with elevated serum PSA (up to 15 ng/mL) or DRE abnormalities, who underwent mpMRI followed by template-mapped biopsy (TPM) and SBx. The procedure was used as a reference test for PCa detection as the gland biopsies were taken every 5 mm. Authors defined clinically significant cancer as Gleason score ≥4 + 3 or any grade with cancer length ≥ 6 mm, which was found in total of 230 patients (40%). MpMRI showed better sensitivity (93% vs. 48%, *p* < 0.05) and negative predictive value (NPV) (89% vs. 74%, *p* < 0.05) than SBx, while SBx was more accurate in terms of specificity (41% vs. 96%, *p* < 0.05) and positive predictive value (PPV) (51% vs. 90%, *p* < 0.05). Authors of PROMIS trial suggested mpMRI as a triage test, which allows safe reduction of performed biopsies by approximately 25%. However, standalone mpMRI has the limitations due to its low specificity (41%).

Several recent studies have demonstrated the advantage of FBx over SBx in the detection of csPCa [11,26,27,28,29,30,31]. In 2018, Kasivisvanathan et al. reported the randomized controlled trial of 500 men with PCa suspicion, randomly assigned for FBx or SBx—the PRECISION trial [11]. The authors found FBx to be more relevant in PCa detection than SBx, showing csPCa in 38% of FBx compared with only 26% of SBx (*p* = 0.005). Moreover, there was 13% less insignificant PCa detected with FBx. However, the promising results of the study had one important limitation. The group of 71 patients who had negative mpMRI imaging (PI-RADS < 3) did not undergo any biopsy—mpMRI is known to have 88–89% NPV for csPCa, making the results unreliable [7,32]. Bass et al., in 2021, published the meta-analysis of 40 studies (8456 men) in which FBx and SBx were compared [30]. Authors used cancer detection rate (CDR) as the main accuracy measure but calculated by dividing the number of patients with detected PCa by FBx (or SBx), with the total number of detected PCa by FBx and SBx together (instead of the number of patients with cancer detected by specific method, divided by the total number of patients, which is used as “CDR” for the rest of this article). The significant difference in the CDR in favor of FBx was established (0.83 [95% CI 0.76–0.90] vs. 0.63 [95% CI 0.53–0.74]). Furthermore, FBx resulted in a lower CDR for insignificant PCa: the diagnostic yield of FBx was 0.08 (95% CI 0.06–0.11), while the yield for SBx was 0.15 (95% CI 0.12–0.17). In randomized controlled trial of 212 biopsy-naïve men, Porpiglia et al. compared FBx and SBx [33]. Out of 107 patients (FBx group) mpMRI was positive in 81 (75.7%) who underwent FBx. Men assigned to the FBx group with negative mpMRI findings underwent SBx (false negative check). The SBx group consisted of 105 patients who underwent SBx without pre-mpMRI. Significant differences between FBx and SBx in the overall PCa detection rate (60.5% vs. 29.5%, *p* < 0.001) and clinically significant PCa detection rate (56.8% vs. 18.1%, *p* < 0.001) were noted. In 3.8% of patients who underwent SBx after negative mpMRI csPCa was found (+15.4% had insignificant disease). The low probability chance for missing csPCa in mpMRI opens the utility of AS, avoiding unnecessary biopsies. Several researchers reported growing impact of mpMRI in AS [34,35,36,37,38]. Baccaglini et al. in a systematic review and meta-analysis of six studies (741 men) suggested that FBx is overall a better choice than SBx for patients classified for biopsy during AS—the pooled sensitivity for the two methods was 0.79 (95% CI 0.74–0.83) and 0.67 (95% CI 0.63–0.74), respectively [34]. In another systematic review, Schoots et al. demonstrated that 70% of men qualified for AS have a positive mpMRI, finding it the tool of choice for these patients [36].

In addition to worse results, SBx is performed somewhat “blindly”, with the need to puncture the entire prostate, causing more side effects and pain to patient [39]. Therefore, it would seem logical to fully replace it with FBx. However, it is hard to admit that it would be a proper maneuver. While the replacement of SBx with FBx is theoretically possible, the complete removal of SBx from PCa diagnostics is controversial, due to false negative results of mpMRI findings. Indeed, multiple studies suggests that FBx is individually superior to SBx, yet a combined approach was found to be the most effective one (Table 1) [26,40,41,42,43,44,45]. Such conclusion was confirmed by Ahdoot et al. who, based on a prospective study of 2103 men, found combined biopsy to diagnose csPCa in 208 more men (9.9%) than either SBx or FBx alone [26]. The cohort was divided into three groups based on the biopsy Gleason grade: group 1—ISUP grade group 1, group 2—ISUP grade group 2 or 3, and group 3—ISUP grade group 4 or 5. The International Society of Urological Pathology (ISUP) PCa grading system, which is used to assign patients to the appropriate risk group, is demonstrated in Table 2 [46]. The authors described the histopathological analysis of surgical specimens from the patients who underwent subsequent radical prostatectomy (404 men), demonstrating the lowest percentage of up-grades to group 3 in combined biopsy (3.5%), as compared with FBx (8.7%) and SBx (16.8%). Similar results were provided by Filson et al. in a prospective study of 825 men with positive mpMRI findings. All patients underwent both FBx and SBx identifying csPCa in a total of 289 (35%) cases with combined approach, while only 229 (28%) cases with FBx alone and 199 (24%) cases with SBx alone [40]. Another prospective study of Elkhoury et al. demonstrated that performing FBx alone would miss 19% of csPCa cases [41]. By comparison, performing only SBx in every patient would miss 18% of such cases. Nevertheless, some csPCa that is missed by FBx is detected by SBx and vice versa, which resulted in total CDR of 70% for a combination of these methods. Rouviere et al. published a prospective multicenter paired-cohort study of 251 men suspected of PCa who underwent mpMRI. In case of positive mpMRI findings, patients received FBx in addition to SBx, which was performed on all patients by a separate urologist. csPCa was diagnosed in total of 37% of patients, while standalone FBx and SBx would miss 5.2% (95% CI 2.8–8.7) and 7.6% (95% CI 4.6–11.6) of those cases, respectively [42]. Researchers agreed to remain the combination of FBx and SBx in the diagnostic pathway of PCa, as the current standard, however, it needs further studies on alternative methods.

## 4. Ultrasound Techniques in the Prostate Cancer Diagnostics

The main target of implementing ultrasound techniques to PCa diagnostic pathway is to overcome the limitations of mpMRI, such as high costs, availability, long procedure time, or potential exclusion of patients with claustrophobia, renal failure, pelvic hardware, or cardiac implants. A biopsy strategy based on targeting under real-time visualization, rather than relying on mpMRI in FBx, would significantly simplify the entire diagnostic process. Therefore, it would be satisfactory if the proposed ultrasound technique achieved results comparable to mpMRI, as it would not have such limitations. The development of ultrasound techniques could possibly lead to a full replacement of mpMRI in the diagnostics of PCa, both in a biopsy qualification and as a targeted biopsy guidance tool. Moreover, ultrasound techniques may become an addition to the current guidelines, which will increase the diagnostic accuracy of mpMRI. Notwithstanding the wide diversity of ultrasonography devices and their common availability, some hindrances emerge. First of all, finding a physician with an sufficient experience in novel ultrasound modalities is very rare. Despite comprehensive utilization and increasing quality of the techniques, there is no significant amount of research on their role in PCa diagnostics. It may lead to some diagnostic biases, which would be less likely to occur in very well studied mpMRI and TRUS-guided biopsy. This clearly indicates the need for consecutive research and obtaining the results of comparative studies. Examples of classic TRUS, which is commonly used ultrasound technique in prostate imaging, are demonstrated in Figure 1.

### 4.1. High Frequency Micro-Ultrasound

Micro-ultrasound scanners operate at higher frequency (29 MHz) traditional TRUS scanners (5–12 MHz). It utilizes a new angled side-fire sagittal transducer which enables 300% higher resolution at the cost of reducing wave penetration depth to 6 cm. As with traditional TRUS, it provides real time, transrectal imaging of ROIs but with a higher quality. MicroUS allows to detect additional focal lesions, “hidden” from TRUS. However, the reduced wave penetration may underperform in very large prostates or anterior located lesions. It is worth noting that prostate size was recently found to have a strong inverse correlation with the incidence and aggressiveness of PCa proven on TRUS-guided biopsies [47]. No study demonstrated contrary results. This indicates that large prostates may be protective of PCa when compared to smaller prostates, which is favorable for the limited wave penetration of MicroUS. This finding is supported by the study of Lophatananon et al., who enrolled 2767 men suspected of PCa and found mean gland volume higher in men with a benign diagnosis (68.1 mL, SD  =  35.5, *p*  <  0.0001) compared to any PCa (52.5 mL, SD = 29.0, *p*  <  0.0001) or csPCa (51.9 mL, SD  =  30.0, *p*  <  0.0001) diagnosis [48]. However, the same small PCa lesion is easier to be hit by a biopsy needle in a small prostate than in a much larger prostate. Nevertheless, the impact of prostate size should be taken into account in the future investigations of prostate imaging techniques to better understand this phenomenon. The first comparison of MicroUS and TRUS was demonstrated by Pavlovich et al. in 2014 [49]. The authors utilized only a 21 MHz micro-ultrasound scanner, which did not prevent obtaining very promising results—both sensitivity (65.2% vs. 37.7%) and specificity (71.6% vs. 65.4%) were improved. The new technique quickly caught the attention of researchers. As the first part of a recent multi-institutional randomized controlled trial of 1676 men with PCa suspicion (clinically qualified for biopsy), a standardized, upgraded protocol, based on the 29 MHz ExactVu™ system (Exact Imaging, Markham, ON, Canada)—PRI-MUS (prostate risk identification using micro-ultrasound) protocol was established [50,51]. PRI-MUS, similar to PI-RADS for mpMRI, consists of 5-point grading system, from 1 for very likely benign to 5 for very likely malignant (Figure 2). In this trial, patients were randomly assigned to traditional SBx or MicroUS-guided biopsy with the first generation 21 MHz device. The implementation of newly developed PRI-MUS, with provided training for investigators in real-time targeting ROIs, in the middle of the trial drastically improved sensitivity in the MicroUS arm (60.8% vs. 24.6% without PRI-MUS, *p* < 0.01), while reducing specificity (63.2% vs. 84.2% without PRI-MUS, *p* < 0.01). Overall, the detection of csPCa was not found to be different in each arm. However, the MicroUS arm included patients examined before and after the PRI-MUS implementation. It was stated that the detection of csPCa improved from 32% without PRI-MUS to 39% with PRI-MUS (*p* < 0.03) and it was improving as the experience of the investigators increased. This may lead to the conclusion that MicroUS is, indeed, superior to traditional TRUS as a biopsy guidance tool, but requires experienced investigators and the use of PRI-MUS protocol. Abouassaly et al. described the first 8 months of using the second generation 29 MHz MicroUS in place of TRUS (SBx) at the Clevelend Clinic [52]. The results of this prospective trial seems to confirm the conclusion of MicroUS superiority—out of 67 enrolled patients who underwent both real-time targeted MicroUS-guided biopsy (MicroUS-Bx) and SBx, 21 were diagnosed with csPCa. MicroUS-Bx detected 100% of the cases, when SBx missed six of these. While very promising, these were just the early results from a small, single-center study.

The question is how MicroUS-Bx performs in a comparison with FBx—which is the current gold-standard addition to SBx, but requires mpMRI with all its flaws. There are multiple recent studies which compare these two types of biopsies [52,53,54,55,56,57,58,59,60,61]. The summarized results of the studies are shown in Table 3. In the first prospective trial, Astobieta Odriozola et al. performed MicroUS-Bx and FBx in 35 patients with clinical PCa suspicion [53]. Both targeted biopsies were taken during the same procedure after systematic sampling—first, a urologist, who was blinded to the mpMRI report, performed MicroUS-Bx, and then mpMRI targets were acquired using cognitive fusion. Overall, 21 patients were found to have csPCa of which FBx detected 12 (57%), while MicroUS-Bx 20 (95%)—one patient was detected only in the SBx. Claros et al. retrospectively compared two cohorts; the first consisted of 222 patients undergoing FBx, and the second consisted of 47 patients undergoing MicroUS-Bx [56]. The CDRs for csPCa were significantly different with values of 23% and 38% for FBx and MicroUS-Bx (*p* = 0.02), respectively. Cornud et al. reported very optimistic results of the prospective study of 118 men [58]. A total of 144 ROIs, including 114 (79%) mpMRI+/MicroUS+ ROIs, 13 (9%) mpMRI+/MicroUS− ROIs, and 17 (12%) mpMRI−/MicroUS+ ROIs, were sampled and analyzed. CsPCa was found in 70/114 (61%) mpMRI+/MicroUS+ ROIs, in 0/13 (0%) mpMRI+/MicroUS− ROIs, and in 4/17 (24%) mpMRI−/MicroUS+ ROIs. MicroUS-Bx demonstrated perfect sensitivity of 100%, while FBx missed four csPCa cases, decreasing the sensitivity to 94%. However, firstly, both procedures were performed by the same urologist, who was not blinded to the results of either one and, secondly, only patients with PI-RADS ≥ 3 were included in the analysis, which may be associated with a risk in selection bias. Another study was demonstrated by Lughezzani et al. on a bigger cohort of 320 men [61]. The inclusion criterion also was the presence of at least one PI-RADS ≥ 3 lesion at mpMRI. Nevertheless, in this study, separate urologists performed FBx and MicroUS-Bx, who were blinded to the results of the other procedure. Additionally, each patient underwent systematic sampling for the best possible evaluation of PCa. Overall, 116/320 (36.3%) patients were diagnosed with csPCa by any method. MicroUS was associated with high sensitivity (90%) and NPV (82%), while the specificity (26%) and PPV (41%) were lower. In total, 27/116 (23%) cases were identified only by targeted sampling. Among them, 21 patients were positive in both MicroUS-Bx and FBx, 3 were only MicroUS-Bx-positive, and 3 were only FBx-positive. On the other hand, 12/116 (10%) csPCa cases were missed detected only by SBx. A similar study was performed by Wiemer et al., but there was no inclusion criterion of PI-RADS score [59]. In a cohort consisted of 159 men, 78 (49%) were diagnosed with csPCa. MicroUS-Bx identified 74/78 csPCa cases (95% sensitivity), while FBx identified 55/78 of the cases (71% sensitivity). Moreover, PPV on a lesion level was 41% and 30% for MicroUS and mpMRI, respectively (*p* = 0.02). Rodriguez Socarras et al. assessed PCa diagnosis accuracy of transperineal approach for MicroUS-Bx and FBx [57]. A total of 194 men with clinical PCa suspicion, with or without mpMRI findings (35 patients had PI-RADS ≤ 2), underwent MicroUS-Bx, FBx and SBx in the same procedure. 81 patients in total were found to have csPCa. Of them, 11 (14%) cases were diagnosed only by MicroUS-Bx. The detection sensitivities for csPCa were 99% and 84% for MicroUS-Bx and FBx, respectively. No infection or fever was observed. The results indicate that transperineal MicroUS-Bx is safe and offers good accuracy, but should be validated in future trials.

There are two major limitations in most of the aforementioned studies. The first is the inclusion criterion of PI-RADS score ≥ 3 at mpMRI. For accurate comparison of MicroUS-Bx and FBx the cohort should not be pre-selected based on mpMRI findings. The second limitation is that these studies consisted of only single-center cohorts. Finally, Klotz et al. overcame the limitations with the first prospective multicenter study of 1040 men from 7 countries [60]. The study included men with clinical PCa suspicion, regardless of the mpMRI results. The biopsy procedure consisted of 2–3 cores from each MicroUS or mpMRI ROI and 12–14 systematic samples. However, in only 2/11 centers, the urologists who performed MicroUS-Bx were blinded to mpMRI findings, and the biopsy protocol differed somewhat between the involved centers. In a total of 877/1040 (84%) and 864/1040 (83%) patients, ROI was detected using MicroUS and mpMRI, respectively. Overall, csPCa was diagnosed in 411/1040 (40%) men. MicroUS demonstrated higher sensitivity than mpMRI (94% vs. 90%, respectively) and lower specificity (22% vs. 23%, respectively). The NPV was significantly higher for MicroUS (85% vs. 77%), the false positive rate and PPV were similar for both methods. These good and promising results are in line with two published meta-analyzes in 2021 of 1125 and 1081 men [62,63]. Both studies demonstrated comparable detection rates of csPCa for MicroUS-Bx and FBx, and the authors consider MicroUS as an attractive alternative to mpMRI in PCa diagnostic pathway. Moreover, MicroUS can be potentially capable to estimate the presence of csPCa in patients with equivocal mpMRI findings (PI-RADS = 3) [64]. Currently, the most anticipated study is the OPTIMUM trial—a 3-arm randomized controlled multicenter trial which will comprehensively compare MicroUS-Bx, FBx and biopsy guided by simplified mpMRI/MicroUS “contour-less” fusion in the detection of csPCa [65]. OPTIMUM aims to determine whether MicroUS can replace mpMRI in the diagnostics of PCa or if it can enhance the performance of currently used methods as an additive tool. Additional studies are warranted for further evaluation of this promising technology. An example of MicroUS image interpretation using PRI-MUS protocol is demonstrated in the Figure 3.

### 4.2. Contrast-Enhanced Ultrasound

CEUS is another relatively new technique which utilizes intravenously injected microbubbles as the ultrasound contrast agents to enhance the vascular signals. The microbubbles contain low solubility gases (e.g., perfluoropropane, perfluorocarbon, or sulfur hexafluoride) closed in flexible shells of phospholipids or liposomes with a diameter of approximately 3 to 5 μm, which is slightly smaller than red blood cell. The most common CEUS contrast is SonoVue^®^ (Bracco, Milan, Italy), which utilizes sulfur hexafluoride [66]. The safety of SonoVue^®^ is well documented and its tolerance in clinical practice is excellent, especially when compared with iodinated and gadolinium contrasts [67,68]. The microbubbles amplify the backscatter of ultrasound waves, resulting in the intensified signals from the blood flow. This method is commonly used in well vascularized organs like liver or kidneys [69,70]. Nevertheless, it can be performed off-label for the assessment of PCa (despite lower perfusion in the prostate than in than in the mentioned organs), due to the confirmed correlation between angiogenesis and the presence of PCa, its stage and survival [69,71]. CEUS was found to have the ability to demonstrate the asymmetric and intense intraprostatic microcirculation-characteristic of PCa, which is beyond the resolution of classic Doppler ultrasound [13]. PCa shows rapid, profuse inflow and outflow, resulting in a faster and stronger enhancement with an earlier wash-out in the venous phase, compared with surrounding benign tissue. However, the wash-out in prostate adenocarcinomas tends to be irregular; therefore, the rapid wash-in is more reliable feature. CEUS may also be helpful in the differentiation of reactive changes, such as prostatitis, which is associated with more spacious hyperenhancement and regular prolonged wash-out. One of the most important limitations of CEUS in PCa diagnostics is the benign prostate metaplasia (BPH), which may obscure the tumor blood flow by increasing the size and vascularization of the transition zone. Furthermore, it might be difficult to detect PCa in the apical and dorsal prostate areas for which endoluminal and 3D CEUS multifrequency probes may be necessary [72]. Lastly, the bolus injection of the microbubble contrast agent provides only temporary enhancement during the intravascular phase, reducing the time to distinguish between PCa and surrounding benign tissue to less than one minute. However, this limitation can be overcome by extending the time of enhancement by infusing the microbubble contrast after bolus injection.

The early studies on CEUS in PCa diagnostics utilized power Doppler ultrasound and Levovist^®^—the first transpulmonary contrast agent registered for radiology procedures. Introducing the microbubble contrast agent quickly augmented the results of prostate biopsies. Roy et al., in one of the first trials of Levovist^®^ in prostate biopsy, reported significant improvement of adding CEUS-targeted biopsy to traditional SBx in both sensitivity (from 54% to 93%) and specificity (from 79% to 87%) [73]. So far, two large clinical trials comparing the effectiveness of real-time targeted CEUS-guided biopsy (CEUS-Bx) and SBx have been performed. In the first one from 2010, Mitterberger et al. retrospectively enrolled 1776 men with PSA ≥ 1.25 ng/mL [74]. Each patient underwent a 5-core CEUS-Bx, and then another investigator, who was blinded to the CEUS findings, performed SBx. PCa was diagnosed in a total of 559/1776 (31%) men, of which CEUS-Bx identified 476/559 (85%) cases, and SBx 410/559 (73%) cases. In total, 149/559 (27%) of cancer-positive cases were detected only by CEUS-Bx, while 83/559 (15%) of such cases were detected only by SBx. The results of the second trial, this time a prospective one, were released in 2019 [17]. In the trial, Yunkai et al. examined 1024 consecutive patients qualified for prostate biopsy based on elevated PSA and/or abnormal DRE. The biopsy procedure included CEUS-Bx with 2–3 cores sampled for each ROI and SBx. The group who performed SBx was blinded to the CEUS results. Overall, prostate biopsy revealed csPCa in 326/1024 (32%) cases. The sensitivities demonstrated by both methods were 90% and 79% for CEUS-Bx and SBx, respectively. CEUS-Bx resulted in 67 cases of csPCa that were missed by SBx. Conversely, SBx detected 32 csPCa cases that were missed by CEUS-Bx. Additionally, SBx was associated with an identification of 58 insignificant PCa cases, while CEUS-Bx with only 12 such cases. However, despite the better results, CEUS-Bx should not be performed without following SBx, especially in patients with low PSA elevation, in whom CEUS-Bx alone may miss significant number of PCa cases. Lu et al. investigated the usefulness of CEUS-Bx and SBx in the three groups of men with PSA values of: 4–10 ng/mL, 10–20 ng/mL, and >20 ng/mL [75]. SBx was found to have significantly higher CDR than CEUS-Bx in PSA 4–10 ng/mL group (45% vs. 33%, *p* = 0.01). The CDRs in PSA 10–20 ng/mL and >20 ng/mL groups were higher for SBx as well, but showed no statistical significance (50% vs. 46%, *p* = 0.15, and 79% vs. 77%, *p* = 0.15, respectively). The results of the studies comparing CEUS-Bx and SBx performance are summarized in Table 4. Furthermore, the meta-analysis of 16 studies (2624 patients) demonstrated diagnostic performance of CEUS in PCa with the pooled sensitivity, pooled specificity, NPV, and PPV of 70, 74, 59, and 82%, respectively [76]. The results revealed that the use of CEUS in the diagnosis of PCa is indeed promising, but not adequately sensitive to be utilized as the sole biopsy guidance tool and cannot completely replace SBx. Nevertheless, the selection of CEUS ROIs for additional targeted sampling significantly enhances the diagnostic accuracy of SBx, and this method should be utilized this way [17,77].

Currently, to improve the conventional two-dimensional (2D) CEUS, the computer-aided quantification of contrast-ultrasound diffusion imaging (CUDI) was demonstrated. CUDI provides several parametric maps of wash-in rate generated from CEUS recordings, based on which the software can automatically estimate the heterogeneity of the enhancement and draw the areas with abnormal enhancement on a 3D model, which can be later utilized as the ROIs for targeted biopsy [78,79]. This method potentially allows a decrease in the user dependency, speed-up of reading, and improved accuracy. Postema et al. compared the abilities of CEUS and CUDI parametric maps to predict the locations of PCa [80]. Each of the 82 consecutive patients underwent SBx resulting in a total of 651 biopsy cores, of which 141 were malignant. CEUS failed to predict the location of 40 cores with csPCa, while CUDI parametric maps would miss csPCa in 23 cores. In the per-prostate analysis the interpretation of CUDI parametric maps was associated with higher sensitivity (91% vs. 73%), PPV (57% vs. 50%), NPV (90% vs. 79%), and slightly lower specificity (56% vs. 58%) than CEUS alone. However, the results cannot be compared with SBx due to the lack of targeted sampling. In the next step, Postema et al. aimed to determine the values of CEUS and CUDI in correlation with radical prostatectomy specimens [81]. This multicenter study included 133 men scheduled for radical prostatectomy with preoperative CEUS imaging and generation of CUDI parametric maps. The 3D models of both imaging and pathology were created and fused. By using automated fusion of the two 3D models, the authors minimalized frequent limitations of the manual correlation of ultrasound images and histopathological specimens, such as plane angulation mismatch and plane selection error. The performances of both CEUS and CUDI were similar with sensitivity of 81% and 83%, respectively, and specificity of 64% and 56%, respectively. Average areas under the receiver operating characteristics (ROC) curve were 78% and 79% for CEUS and CUDI, respectively. However, the disappointing results might be affected by the observers’ considerable amount of prior CEUS experience, and lack of it with the interpretation of CUDI maps. Recently, Mannaerts et al. prospectively enrolled 142 men suspected of PCa to compare the effectiveness of targeted biopsy based on CUDI software-analyzed parametric maps (CUDI-Bx) and FBx [82]. Overall, 62/142 (43%) patients were diagnosed with csPCa, of which CUDI-Bx identified 40/62 cases (65% sensitivity) and FBx identified 41/62 cases (66% sensitivity). SBx was also performed and had superior results to both targeted approaches identifying 56/62 (90% sensitivity) csPCa cases, which was the reason to stop the trial. Despite the fact CUDI-Bx and FBx csPCa detection rates were similar, FBx demonstrated significantly fewer false-positive findings (18% vs. 53%). All the presented results led to the conclusion that CEUS might not be the best standalone tool for the diagnosis of PCa, but has a good supportive value for the currently used methods. Besides its potential role in the PCa diagnostics, CEUS allows for better monitoring, with the assessment of hypoperfusion and necrosis after radiologic interventional treatment of PCa, for example, irreversible electroporation or focal therapy with high-intensity focused ultrasound [83,84,85]. In addition to the post-treatment use, CEUS is capable of intraoperative assessment of the extent of focal high-intensity focused ultrasound therapy as well as estimating the risk of residual disease [86]. Moreover, it can be utilized in the imaging evaluation of prostatic artery embolization for the treatment of benign prostatic hyperplasia [87]. The study of Jiang et al. demonstrated that CEUS may find the use in the prognosis of PCa [88]. The authors revealed the positive correlation between the degree of enhancement and the aggressiveness of PCa. This finding was later confirmed by Baur et al., but at the same time, the authors found dynamic contrast enhanced MRI to perform better than CEUS in predicting the aggressiveness of PCa [89]. Nevertheless, quantification of CEUS parameters during FBx was able to discriminate PCa aggressiveness in clinical practice [90].

### 4.3. Shear-Wave Elastography

Another promising area in the field of ultrasound diagnostics of PCa is the use of SWE. Bercoff et al. described SWE in 2003 and demonstrated the early results from its first clinical implementation [91,92]. SWE is based on the generation of shear waves in tissue using acoustic radiation force from multiple focused ultrasound beams [93]. The device generates two shear waves which spread within the tissues with velocity variation based on the stiffness of the tissues. The velocity of the waves is higher in stiffer tissues and lower in softer tissues, and measuring the difference enables one to achieve a dynamic quantitative color map which reflects the tissue stiffness. The map is an overlay of the color-coded interpretation of the shear wave velocity (m/s) converted into Young’s modulus (expressed in kilopascals [kPa]) on the ultrasound imaging in real time, with red presenting stiff tissues and blue presenting soft tissues (Figure 4). The utilization of SWE in prostate examination is relatively new and is possible thanks to the introduction of shear-wave endocavitary transducers. Prostate SWE examination has to be performed with an appropriately selected elasticity scale of 70–90 kPa. The signal must achieve proper stabilization to minimize imaging artifacts, which requires no compression of the prostate and rectal wall. Usually, if the prostate is not very large, SWE can scan the entire gland from base to apex. The transitional zone of the healthy prostate glands without BPH is soft, with elasticity values up to 30 kPa, while as BPH progresses, it becomes stiffer, with elasticity values between 30 and 180 kPa, presented as heterogenous red pattern [94]. The peripheral and central zones remain soft despite the development of BPH, with elasticity values ranging from 15 to 25 kPa, presented as homogenous blue pattern. PCa was demonstrated to have a high stiffness at the median value of 91 kPa and is usually strongly color coded in red [95]. However, not all PCa are stiff and, conversely, not all stiff lesions are PCa, which is the main limitation of SWE. Another limitation is an examination of very large prostates in which, firstly, the gland protrudes toward the rectal wall causing tissue compression and artifacts; secondly, SWE tissue penetration depth is restricted to 3–4 cm, and is not able to cover the entire prostate, causing some anterior lesions to potentially remain hidden [96]. Nevertheless, most PCa are located in the peripheral zone, which is reachable in the overwhelming majority of the cases [97]. Furthermore, multiple, or large prostate calcifications may limit the performance of SWE, due to their extremely high stiffness. Although it is known that performing SWE is associated with a learning curve, there is no established standardized training procedure so far. Moreover, investigators need to pay attention to the fact that tissue stiffness is influenced by the imaging plane—tissues have higher elasticity values on sagittal than on axial imaging [98]. Therefore, prostate SWE should always be performed in the axial plane to obtain the most representative values of Young’s modulus.

The World Federation for Ultrasound in Medicine and Biology (WFUMB) has developed guidelines for the use of SWE, including basic science, breast, liver, and thyroid elastography [99,100,101,102]. Recently, for the first time, WFUMB introduced guidelines for the utilization of SWE in PCa diagnostics [94]. According to the guidelines, a stiffness value greater than 35 kPa is suggestive of a malignancy and a reason to perform biopsy. This value is based on findings of Barr et al. who reviewed 318 biopsy cores but found PCa in only 26 of those cores [96]. The 35 kPa cut-off is in line with the study of Correas et al. who investigated a total of 1040 peripheral zone sextants of which 129 contained PCa tissue [103]. The sensitivity, specificity, PPV, and NPV for differentiating benign from malignant lesions were 96, 85, 48, and 99%, respectively. However, Boehm et al. reevaluated the cut-off suggesting that the CDR for the currently used threshold is unsatisfying [95]. The authors found the most informative elasticity value for the prediction of PCa as greater than 50 kPa. This cut-off value was associated with the best balance between sensitivity (81%) and specificity (69%), with 67% PPV and 82% NPV. In the most recent study, Ji et al. proposed an even higher optimal threshold of 62.27 kPa, which was characterized by 81% sensitivity and 75% specificity [104]. Moreover, the authors revealed a positive correlation between Young’s modulus and PCa aggressiveness, which can be utilized in the cancer prognosis and treatment. With SWE providing a real-time detection of ROIs, the value of real-time targeted SWE-guided biopsies (SWE-Bx) was investigated. In the meta-analysis comparing SWE-Bx and SBx, Tu et al. analyzed seven studies including five cohorts and two randomized controlled trials [105]. In the five cohorts (a total of 698 men with PCa suspicion), SWE-Bx did not outperform SBx (CDR 70% vs. 81%, respectively). Nevertheless, the two analyzed randomized controlled trials demonstrated a more favorable trend towards greater CDR when a combination of both SBx and SWE-Bx was performed than when SBx alone was performed (46% vs. 40%, respectively). Further study about SWE-Bx additive value to SBx was demonstrated by Zhang et al. in the prospective study of 489 consecutive men suspected of PCa who underwent SWE-Bx and SBx [106]. PCa was detected by both methods in a total of 221/489 (45%) patients, while SBx alone was associated with significantly lower detection rate of 33% (*p* < 0.05), and only 162/221 (73%) cancer-positive patients disclosed. It resulted in the recommendation in WFUMB guidelines stating that SWE-Bx should always be performed with standard SBx, enhancing the results of SBx [94]. By the date of publication of WFUMB guidelines in 2017, there was no studies to compare SWE with mpMRI in the diagnosis of PCa. Therefore, in the document, there is no recommendation about performing SWE instead of mpMRI or as an addition to mpMRI. The status of insufficient evidence has changed recently; two meta-analyzes of SWE performance in the detection of PCa were demonstrated, with the results comparable to those seen in mpMRI. The first meta-analysis included eight studies (a total of 1028 patients), and the pooled sensitivity was 83% (95% CI, 0.66–0.92) with the pooled specificity of 85% (95% CI, 0.78–0.90) [14]. The second one included nine studies, resulting in even better pooled sensitivity and specificity (86% [95% CI 0.75–0.92] and 89% [95% CI 0.82–0.93], respectively) [15]. Moreover, the pooled area under the ROC curve of 94% (95% CI 0.91–0.95) suggested an overall good accuracy of SWE. However, there is still lack of clinical trials with head-to-head comparison of these two imaging modalities. Recently, the development of 3D prostate SWE enabled multiplanar reconstruction and potentially more accurate guiding capabilities for targeted biopsies. Shoji et al. evaluated the efficiency of 3D SWE in the detection of PCa for the first time in clinical practice [107]. This preliminary study prospectively recruited 10 patients with elevated PSA and suspicious mpMRI findings. Targeted biopsies from mpMRI ROIs were initially performed and followed by SBx. Each biopsy punctured lesion was examined by 3D SWE with the measurement of Young’s modulus. Then the pathological biopsy results were compared with the images acquired from mpMRI and 3D SWE. The cut-off value of Young’s modulus was established on the basis of the ROC analysis at 41 kPa. The cut-off was associated with PCa detection sensitivity, specificity, PPV, and NPV of 58, 97, 86, and 87%, respectively. By combining the cut-off value of Young’s modulus with PI-RADS score, PCa was correctly identified in 21 of 23 lesions (91%), and the sensitivity, specificity, PPV, and NPV of PCa detection improved to 70, 98, 91, and 92%, respectively. This new modality may help to achieve better accuracy in predicting ROIs than currently used mpMRI alone.

### 4.4. Multiparametric Ultrasound

As CEUS and SWE have both demonstrated promising results, a prospective randomized comparison of these modalities was performed [108]. As neither CEUS-Bx nor SWE-Bx are sensitive enough to be performed without SBx, the study investigated their accuracy as an addition to SBx. A total of 52 patients with PCa suspicion were randomly assigned to the CEUS-Bx and SWE-Bx groups. After one examiner performed targeted biopsy from ROIs found in CEUS or SWE, the second examiner, blinded to the obtained imaging results, performed SBx. The per core analysis revealed better additive value of SWE-Bx than CEUS-Bx. In the SWE-Bx group, the core-based CDR significantly increased in SWE-Bx cores compared with SBx cores (from 4.5% to 13%, *p* < 0.01), while in the CEUS-Bx group the core-based CDR did not statistically differ (18.8% and 18.3% for CEUS-Bx and SBx, respectively). However, CEUS-Bx was observed to be superior in the mid-gland, while SWE-Bx was better in the apex. Therefore, the idea that CEUS and SWE complement each other resulted in the development of a fusion of these techniques with conventional grayscale and color Doppler ultrasounds, which was called multiparametric ultrasound (mpUS) [109]. Similar to the concept of mpMRI, mpUS is a combination of different methods that benefits from the strengths of each of them to achieve the diagnosis. MpUS examination begins with conventional TRUS transverse and sagittal images in both grayscale and color Doppler. Then, additional ROIs are acquired using SWE and CEUS. Lastly, a map of all mpUS ROIs is created and can be utilized for mpUS targeted biopsy. The technical characteristics of the ultrasound modalities are presented in Table 5. Postema et al. confirmed that combining different ultrasound modalities into mpUS significantly improves the individual performance of these modalities [110]. MpUS was found to have promising potential for the development of focal therapy where non-invasive, precise imaging techniques are important [111,112]. Furthermore, combined performance of mpUS in the diagnosis of PCa was compared with mpMRI. Recently, Zhang et al. prospectively enrolled 78 men who underwent mpMRI, mpUS and then SBx [113]. The targeted samples were not taken. The study was focused on the diagnosis of localized PCa; therefore, the performance of mpUS in the detection of PCa at variable stages were outside the scope of this trial. Nevertheless, the obtained results were very promising, with higher sensitivity, specificity, PPV, NPV, and accuracy than mpMRI (97.4% vs. 94.7%, 77.5% vs. 60.0%, 80.4% vs. 69.2%, 96.9% vs. 92.3%, and 87.2% vs. 76.9%, respectively) for detecting localized PCa. The area under the ROC curve for mpUS was 0.874 ± 0.043 (95% CI 0.790–0.959), which was higher than for mpMRI (0.774 ± 0.055 [95% CI 0.666–0.881]). None of the conventional TRUS, CEUS, and SWE methods alone provided as good results as mpUS. Most importantly, these results are accomplished at lower cost and patient complications than with mpMRI. In another recent study Pepe et al. aimed to evaluate the accuracy of mpUS in the detection of the ROIs found by mpMRI [114]. Sixty patients suspected of PCa underwent mpMRI and mpUS before FBx. MpUS was positive in only 13/21 (62%) cases where mpMRI detected csPCa, and the authors concluded that the additional use of mpUS do not improve mpMRI findings. In a similar study, Drudi et al. acquired comparable results [115]. After performing mpMRI and all mpUS examinations, 82 men with PCa suspicion underwent FBx and SBx. The performances of each of the mpUS modalities were demonstrated individually, and none of them yielded results as good as mpMRI. From the ultrasound methods, SWE was associated with the best sensitivity (85%) and accuracy (77%), but these were both significantly better for mpMRI (96% and 93%, respectively). However, in this study the pooled performance of all mpUS modalities was not presented. Moreover, an important limitation of the results obtained by both Pepe et al. and Drudi et al. is that only the mpMRI ROIs were used for the targeted biopsies, and no targeted samples were taken separately for the mpUS ROIs. Therefore, the studies should not be utilized as a strong opposition to the satisfactory results obtained by Zhang et al. who used, maybe not ideal, but fair for both methods, systematic sampling as the reference. Thus far, there is still a lack of studies performing a direct pathological analysis of ROIs found in mpUS. Certainly, a trial that would compare head-to-head the effectiveness of FBx and mpUS-targeted biopsy is needed.

## 5. Future Perspectives

Based on the aforementioned findings, the novel ultrasound techniques are capable of improving the currently used methods and should be considered as the future direction of PCa diagnostics. These techniques are developing dynamically, and in the near future multiple new technologies will be implemented into clinical practice.

The combination of different imaging modalities is commonly known, with the fusion of classic TRUS and mpMRI being recommended guidance tool for prostate biopsies. Recently, the fusion of elastography and mpMRI was introduced and retrospectively evaluated by Ding et al. [116]. Based on a cohort of 62 men the prognostic performance of the elastographic Q-analysis score combined with PI-RADS for malignancy risk stratification in prostate ROIs acquired from elastography-MRI fusion imaging was investigated. Both elastography and mpMRI separately yielded accurate results with the sensitivities, specificities, PPVs, NPVs, and areas under the ROC curve of 86% and 83%, 82% and 70%, 81% and 71%, 87% and 82%, and 86% and 84%, respectively. Nevertheless, the fusion of these imaging modalities was associated with even higher sensitivity (97%), specificity (88%), PPV (85%), and NPV (95%). Furthermore, the same researchers developed and then validated a nomogram combining the elastographic Q analysis score, PI-RADS score, and clinical parameters for the stratification of patients with PCa [117]. The areas under the curve for predicting csPCa in the training cohort (*n* = 271) and in the validation cohort (104) were 0.936 (95% CI 0.906–0.965) and 0.971 (95% CI 0.9331–1), respectively. With a fusion of mpMRI and novel ultrasound modalities, improvements of currently used methods can be achieved. Therefore, further studies investigating this technology are required.

Another innovation is an introduction of the new microbubble contrast agents used in CEUS. The microbubbles are decorated with site-specific ligands, like antibodies or peptides, to react with the receptors of a specific marker [118]. The most frequently studied markers for cancer imaging are related to tumor angiogenesis, such as the vascular endothelial growth factor receptor-2 (VEGFR2) and α_v_β_3_ integrin—a member of the integrin family, the heterodimer transmembrane glycoproteins. This “targeted” imaging modality is called molecular CEUS and seems very promising at present [119]. However, the addition of new protein antigens to the microbubble contrast was associated with a high immunogenic potential; therefore, they used to be not allowed in humans [120]. In preclinical studies of molecular contrast agents, VEGFR2 targeting was utilized with a good accuracy in the angiogenesis-based detection of PCa in the rat models [121]. Recently, a new technique for the preparation of targeted contrast agents was developed, resulting in the first molecularly targeted ultrasound contrast agent approved for clinical trials. The contrast is called BR55 and has a high binding capability to human VEGFR2. Smeenge et al. demonstrated the first application of BR55 in humans [122]. In this phase 0 study, the feasibility and safety of BR55 for the detection of PCa in patients was investigated. The new contrast agent yielded a good safety profile as no serious adverse events occurred. ROIs were identified with high accuracy and the correlation with histopathological findings was satisfactory. The α_v_β_3_ integrin is the second currently investigated marker for the detection of PCa with molecular CEUS. Several preclinical studies successfully utilized α_v_β_3_ integrin-targeted microbubble contrast for the cancer detection in the breast, ovary, liver, and prostate [123,124,125,126,127,128]. The permission to use molecular CEUS in humans should stimulate further research on the exploration of this promising imaging modality.

With the emerging potential of AI and deep learning methods in medical practice, researchers investigated new possibilities for interpreting images of the prostate. Encouraging results have been reported on AI correctly classifying diseases in various organs based on CEUS images [129]. In the prostate, Wildeboer et al. found that the machine learning analysis of 13 combined parameters related to CEUS perfusion and CUDI dispersion improved the accuracy of PCa localization [130]. Furthermore, the utilization of AI and deep learning framework was demonstrated in the interpretation of molecular CEUS images. The recently invented anti-PSMA (prostate specific membrane antigen) microbubble contrast agent targeted to PCa cells was used in the mouse models of PCa [131]. Two groups of mice were injected with the targeted anti-PSMA contrast agent or the blank contrast agent. CEUS images were acquired and then analyzed using AI. The sensitivity, specificity and accuracy were 10%, 6%, and 7% higher, respectively, for molecular CEUS than non-targeted CEUS. The deep learning framework interpretation of the molecular CEUS images achieved 83% sensitivity, 91% specificity, and 90% accuracy, which exceeds the CEUS performance reported in the literature. Recently, a deep learning system was utilized to fuse and analyze mpUS imaging modalities [132]. The study comprised 50 men with confirmed PCa that were referred for radical prostatectomy. By using all fused mpUS parameters, the deep learning framework outperformed every single parameter analyzed alone. The AI multiparametric analysis reached a region-wise area under the ROC curve of 0.75 and 0.90 for PCa and csPCa, respectively. More powerful deep learning algorithms would allow to combine more parameters, including laboratory and clinical data. It is expected that further advances in ultrasound technology and use of AI will soon enable effective implementation of mpUS in clinical practice [133]. Especially with the development of image fusion techniques and 3D ultrasound models, the deep learning methods may achieve eminent results. An AI system nowadays can be trained to analyze and grade PCa in biopsy samples at a level comparable to that of international experts in prostate pathology, which may soon become the diagnostic standard [134]. Development of a similar system that would detect ROIs in different imaging modalities with the efficiency of experts in prostate radiology should be only a matter of time.

Another area which is currently intensively studied in prostate imaging is the combined utilization of radiomics (such as deep learning framework) and genomics (imaging biomarkers). Genomics information can be explained or decoded by radiomics creating radiogenomics. Over the past few years, the values of different radiogenomics used with MRI have been demonstrated with promising results [135]. However, to date, none of the proposed radiogenomic technologies have been validated in the clinical practice. Moreover, there is still lack of studies investigating radiogenomics potential in the ultrasound PCa diagnostics. Based on the increasing amount of radiogenomics research and the good performance of the ultrasound-based techniques, it is recommended to investigate the potential to combine these technologies.

## Figures and Tables

**Figure 1 cancers-14-01859-f001:**
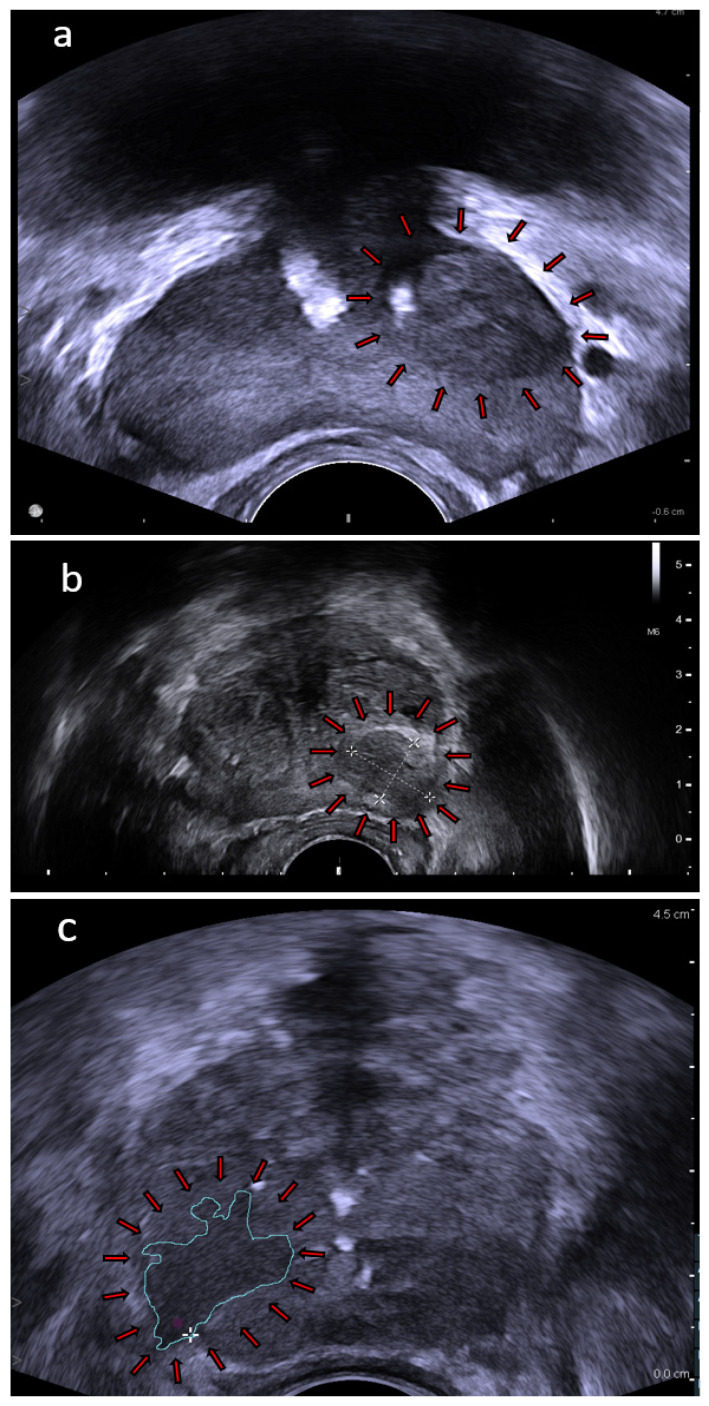
TRUS images: (**a**) prostate with a left side lesion (arrows); (**b**) prostate with a 16 × 12 mm lesion (arrows); and (**c**) prostate with a drawn area of known cancer (arrows). Images provided courtesy of BK Medical.

**Figure 2 cancers-14-01859-f002:**
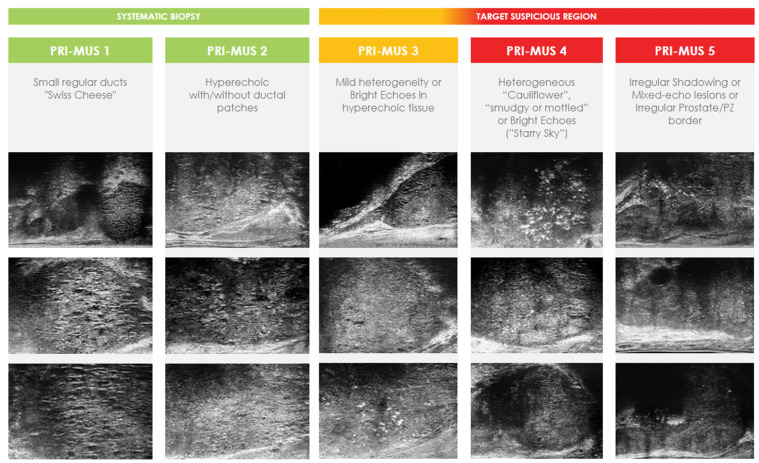
Official examples of ExactVu™ system PRI-MUS grades, from https://www.exactimaging.com (accessed on 17 January 2022); Images provided courtesy of Exact Imaging. Scale bar or magnification.

**Figure 3 cancers-14-01859-f003:**
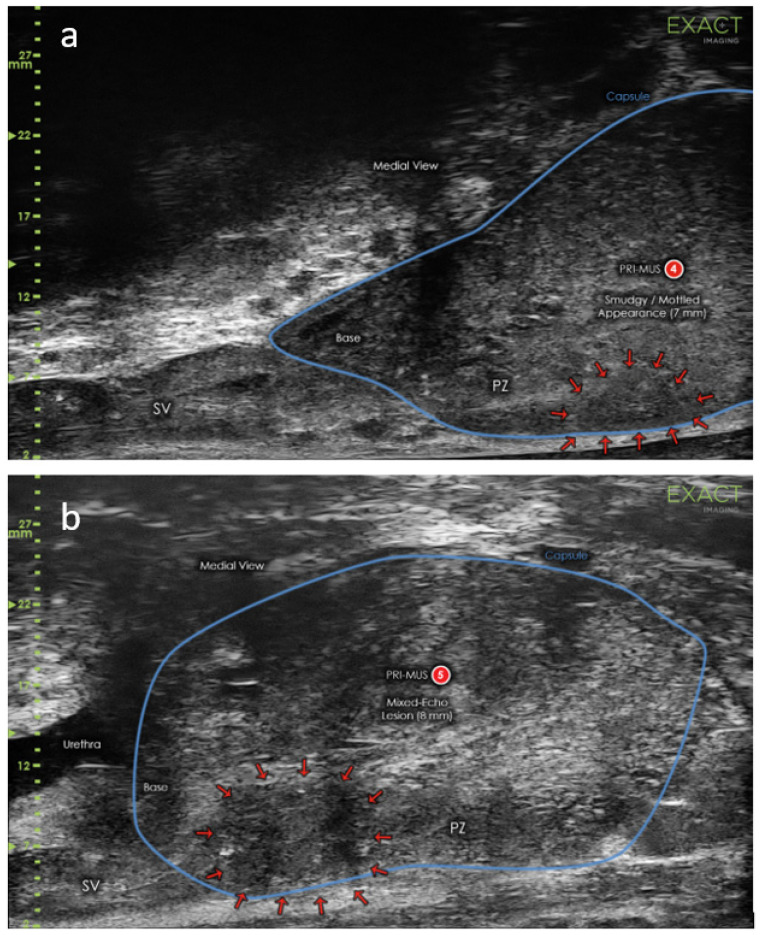
MicroUS images: (**a**) prostate with a PRI-MUS 4 grade lesion (arrows) and (**b**) prostate with PRI-MUS 5 grade lesion (arrows). Images provided courtesy of Exact Imaging.

**Figure 4 cancers-14-01859-f004:**
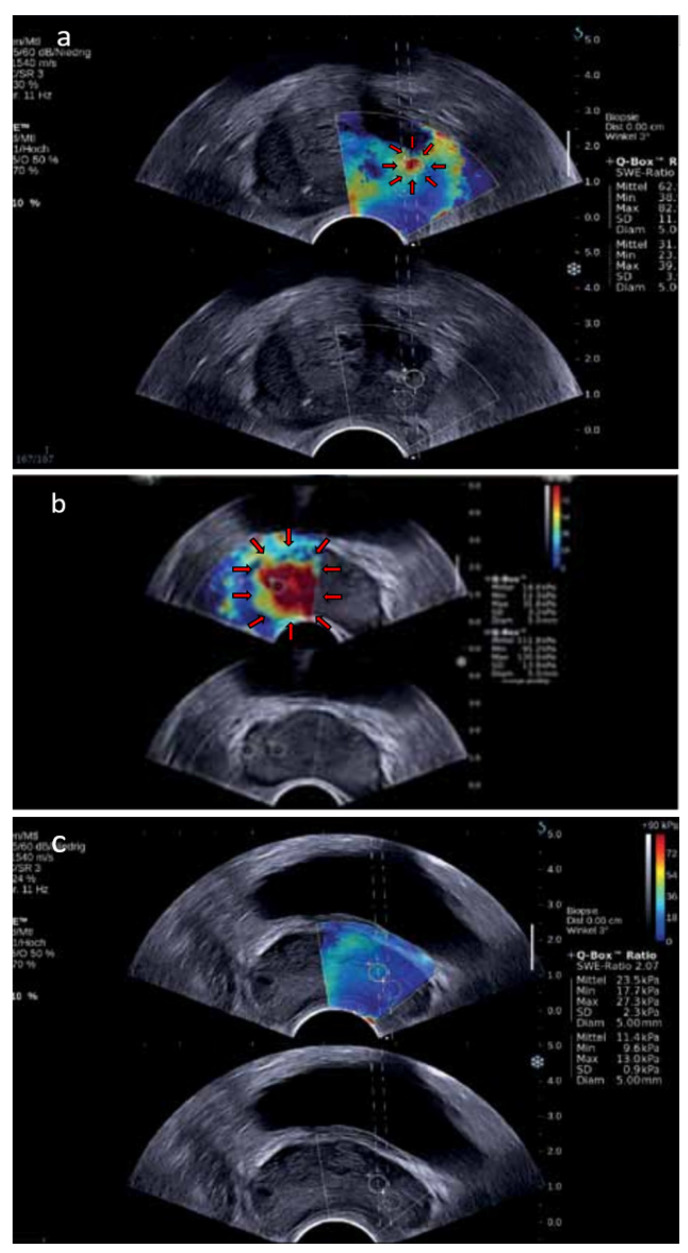
SWE images: (**a**) prostate with a small hard lesion (kPa value: 62) in the peripheral zone (arrows); (**b**) prostate with a huge area of very hard tissue (kPa value: 111), highly suspicious (arrows); and (**c**) prostate with no hard tissue and no evidence of tumor. Images provided courtesy of SONOlife.

**Table 1 cancers-14-01859-t001:** Comparison of cancer detection rates of FBx, SBx, and the two approaches combined.

Study Authors	Year	Number of Patients	FBx CDR	SBx CDR	FBx and SBx Combined CDR
Kasivisvanathan et al. [11]	2018	500	38%	26%	-
Bass et al. * [30]	2021	8456	83% *	63% *	-
Porpiglia et al. [33]	2017	212	60.5%	29.5%	-
Baccaglini et al. [34]	2020	741	31%	30%	-
Ahdoot et al. [26]	2020	2103	52%	53%	62%
Filson et al. [40]	2020	825	28%	24%	35%
Elkhoury et al. [41]	2019	300	62%	60%	70%
Rouviere et al. [42]	2019	251	32%	29%	37%

* The authors used different formula to calculate CDR.

**Table 2 cancers-14-01859-t002:** ISUP PCa grading system.

Risk Group	Grade Group	Gleason Score	Gleason Pattern
Low/Very Low	1	≤6	≤3 + 3
Intermediate (Favorable/Unfavorable)	2 3	7 7	3 + 4 4 + 3
High/Very High	4 5	8 9 or 10	4 + 4, 3 + 5, 5 + 3 5 + 4, 5 + 4 or 5 + 5

**Table 3 cancers-14-01859-t003:** Comparison of MicroUS-Bx and FBx performance in the detection of csPCa.

Study Authors	Year	Number of Patients (MicroUS-Bx/FBx)	Sensitivity MicroUS-Bx/FBx	Specificity MicroUS/mpMRI	CDR MicroUS-Bx/FBx
Astobieta Odriozola et al. [53]	2018	35	95%/57%	40%/91%	57%/34%
Eure et al. [54]	2019	9	89%/56%	x/x	89%/56%
Abouassaly et al. [52]	2020	67/19	95%/80%	x/x	30%/42%
Claros et al. [56]	2020	47/222	x/x	x/x	38%/23%
Klotz et al. [60]	2020	1040	94%/90%	22%/23%	37%/36%
Cornud et al. [58]	2020	118	100%/94%	23%/x	51.4%/46%
Rodriguez Socarras et al. [57]	2020	194	99%/86%	29.3%/x	41%/36%
Lughezzani et al. [61]	2021	320	87%/87%	26%/x	32%/32%
Wiemer et al. [59]	2021	159	95%/71%	15%/x	47%/35%

**Table 4 cancers-14-01859-t004:** Comparison of CEUS-Bx and SBx performance in the detection of csPCa.

Study Authors	Year	Number of Patients	CEUS-Bx Sensitivity	SBx Sensitivity	CEUS-Bx CDR	SBx CDR
Mitterberger et al. [74]	2010	1776	85%	73%	27%	23%
Yunkai et al. [17]	2019	1024	90%	79%	29%	25%
Lu et al. [75]	2021	186	91%	100%	58%	63%

**Table 5 cancers-14-01859-t005:** Main technical characteristics of the ultrasound modalities.

	Imaging Technique	TRUS	MicroUS	SWE	CEUS
Variable					
Wave Type		ultrasound wave	ultrasound wave	shear wave	ultrasound wave
Wave Frequency		5–12 MHz	29 MHz	100–600 Hz	5–12 MHz
Wave Penetration Depth		8–12 cm	6 cm	3–4 cm	8–12 cm
Main Measured Parameter		wave impedance [kg/(m^2^s)]	wave impedance [kg/(m^2^s)]	Young’s modulus (stiffness) [kPa]	perfusion intensity [mL/g]
Guidelines for Prostate Image Interpretation		NCCN Clinical Practice Guidelines in Oncology: Prostate Cancer	PRI-MUS protocol	WFUMB guidelines	-
Contrast Agent		-	-	-	microbubbles

## Data Availability

https://www.exactimaging.com, accessed on 17 January 2022.

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
