# Peer review of "Alternatives for MRI in Prostate Cancer Diagnostics—Review of Current Ultrasound-Based Techniques"

_cancers, 2022, doi:10.3390/cancers14081859_

Round 1

Reviewer 1 Report

The purpose of this review is to present the current role of ultrasound-based techniques in the diagnostic pathway of prostate cancer (PCa). The authors should be congratulated for the work and for addressing an important topic. However, different points warrant mentions:

Comments:

  1. Einspieler et al. (PMID: 28209912) ( DOI: 10.2967/jnumed.116.184457) observed that The detection rate was positively associated to increasing PSA as well as concomitant ADT. I would suggest the authors to perform a subgroup analysis in patients undergoing ADT in order to evaluate this possible variable.
  2. It would appropriate to include the ISUP grade, repeatedly mentioned in the work and not fully described.
  3. I suggest to add a table to resume the different characteristics of the ultrasound-based techniques.
  4. An important limit of the study is that ultrasound-based techniques are really not widespreas and this can became a problem since there are only few physician with an important experience in this regard.
  5. There are few studies about current role of ultrasound-based techniques in the diagnostic pathway of prostate cancer (PCa). Authors shoud discuss it since this can represent a bias.
  6. I believe that authors should include more information also in view of current literature. These interesting papers deserve to read about:

        - https://pubmed.ncbi.nlm.nih.gov/34576134/; DOI: 10.3390/ijms22189971.

 The manuscript is potentially good.

I believe that the study has to be considered for publication on Cancers, although major revisions are required.

Author Response

cancers-1667080

Response letter to the Reviewer #1 Report

We thank the reviewers for their encouraging feedback and appreciate the insightful comments and suggestions.

Below, we provide a point-by-point response to each of the reviewers’ comments.

All changes in the manuscript were highlighted in yellow for clarity.

We hope that the introduced revisions significantly improve the quality of this review and qualify it for further editorial stages.

Sincerely,

Authors

  1. Einspieler et al. (PMID: 28209912) ( DOI: 10.2967/jnumed.116.184457) observed that The detection rate was positively associated to increasing PSA as well as concomitant ADT. I would suggest the authors to perform a subgroup analysis in patients undergoing ADT in order to evaluate this possible variable.

Response: Thank you for your good comment. Indeed, such a correlation has been demonstrated, but no subgroup analysis was performed due to the lack of studies with new ultrasound techniques in such a cohort. Because we present a review-type article, and the Einspieler study using PET / CT has not yet found a reference in the literature on ultrasound techniques, we could not include the proposed analysis in this article

  1. It would appropriate to include the ISUP grade, repeatedly mentioned in the work and not fully described.

Response: Thank you very much for this valuable comment. Indeed, the ISUP scale, introduced in recent years, is widely used, and serves the proper assignment of patients to risk groups. Therefore, we made changes to the manuscript and created Table 2, which describes the ISUP grading.

  1. I suggest to add a table to resume the different characteristics of the ultrasound-based techniques.

Response: Table 5 describing the main technical differences of ultrasound techniques has been added to the manuscript.

  1. An important limit of the study is that ultrasound-based techniques are really not widespreas and this can became a problem since there are only few physician with an important experience in this regard.

Response: A very valuable note. That's why this review was created. To raise the awareness of readers and encourage more widespread using of these ultrasound-based techniques. In the world, there are actually only a few physicians with an important experience in this regard, so despite the availability of apparatuses and the increasing quality of ultrasound methods, we do not have a significant number of studies related to the role of ultrasound in the diagnosis of PCa, i.e. their comparison with the current guidelines. This clearly indicates the need for further research and obtaining the results of comparative works. Appropriate excerpt from the above issues has been added to chapter: 4. Ultrasound Techniques in the Prostate Cancer Diagnostics (lines 218-225)

  1. There are few studies about current role of ultrasound-based techniques in the diagnostic pathway of prostate cancer (PCa). Authors shoud discuss it since this can represent a bias.

Response: Thank you very much, this is a very important remark that is closely related to comment no. 4 and we have included the answer there.

  1. I believe that authors should include more information also in view of current literature. These interesting papers deserve to read about:

        - https://pubmed.ncbi.nlm.nih.gov/34576134/; DOI: 10.3390/ijms22189971.

Response: Thank you. Indeed, this article is very interesting. We have attached to the manuscript data on the possibilities of using radiogenomics in ultrasound techniques presented in the suggested paper, as well as information from other current articles.

Reviewer 2 Report

This paper is presenting an interesting topic on diagnosing clinically significant prostate cancer (PCa) by comparing new imaging technologies, in particular MRI and ultrasound (US).

I have the following questions and recommendations:

  1. The authors do not mention the criteria used in selecting the papers for this review. For a systemic review I recommend specific guidelines (e.g. PRISMA) with defined inclusion and exclusion criteria. Furthermore, it should be mentioned which search engines such as PubMed were used for their review.
  2. The used search engines are also a reflection of the selection bias present in any review.
  3. The descriptions of the different studies in the manuscript are too long, they should be more concise by using more tables with listing of the different studies (such as in Table 1). The lengthy descriptions of the studies are impairing the speed of reading and the line of thoughts. Also, summarizing meta-analyses using Forest plots for the different imaging technologies give the reader a quick overview regarding the clinical value.
  4. The issue of sampling error in biopsy technology should be mentioned. The same small PCa lesion is easier to be hit by a biopsy needle in a small prostate than in a much larger prostate.
  5. As mentioned in the manuscript, prostate size may be a limiting factor for US technologies in large prostates (as outlined in chapter 3.1. on page 6). In this context a recent meta-analysis of US-guided biopsies revealed that 95% of the studies showed a strong inverse correlation between prostate size and the incidence of prostate cancer proven on biopsies (Research and Report in Urology 2021: 13, 749-757). No study could be found showing the contrary. These findings were also supported by a large multi-institutional MRI study in the UK (Lophatananon et al.: Re-evaluating the diagnostic efficacy of PSA as a referral test … in contemporary MRI-based image-guided biopsy pathways. Journal of Clinical Urology, published 12-01-2021). This clinical phenomenon of ‘prostate size matters’ should be mentioned in the manuscript because future studies may consider prostate volume as a parameter in their algorithms.

Author Response

cancers-1667080

Response letter to the Reviewer #2 Report

We thank the reviewers for their encouraging feedback and appreciate the insightful comments and suggestions.

Below, we provide a point-by-point response to each of the reviewers’ comments.

All changes in the manuscript were highlighted in yellow for clarity.

We hope that the introduced revisions significantly improve the quality of this review and qualify it for further editorial stages.

Sincerely,

Authors

  1. The authors do not mention the criteria used in selecting the papers for this review. For a systemic review I recommend specific guidelines (e.g. PRISMA) with defined inclusion and exclusion criteria. Furthermore, it should be mentioned which search engines such as PubMed were used for their review.

Response:  Thank You for this remark. Absolutely valid point. We have added a relevant paragraph (2. Evidence acqusition) describing the methodology, terms of searching for articles and search engines. As it is not a systematic review but a narrative review, it is not possible to apply typical guidelines such as PRISMA.

  1. The used search engines are also a reflection of the selection bias present in any review.

Response:  A fair and important comment. The selection of appropriate literature for a narrative review article is based on the available search engines. Therefore, in order to minimize the bias, we chose the works with the highest level of evidence after careful analysis by the authors

  1. The descriptions of the different studies in the manuscript are too long, they should be more concise by using more tables with listing of the different studies (such as in Table 1). The lengthy descriptions of the studies are impairing the speed of reading and the line of thoughts. Also, summarizing meta-analyses using Forest plots for the different imaging technologies give the reader a quick overview regarding the clinical value.

Response: Thank you, this is a really good suggestion, so we shortened the research descriptions with less important information and included the data from the text in tables, which should improve the transparency of the work. Our article is not a systematic review and a meta-analysis but a narrative review, therefore we could not use tools such as Forest plots in this manuscript.

  1. The issue of sampling error in biopsy technology should be mentioned. The same small PCa lesion is easier to be hit by a biopsy needle in a small prostate than in a much larger prostate.

Response: Thank you very much for paying attention to this issue. It is closely related to the comment No. 5 and the exact answer is there.

  1. As mentioned in the manuscript, prostate size may be a limiting factor for US technologies in large prostates (as outlined in chapter 3.1. on page 6). In this context a recent meta-analysis of US-guided biopsies revealed that 95% of the studies showed a strong inverse correlation between prostate size and the incidence of prostate cancer proven on biopsies (Research and Report in Urology 2021: 13, 749-757). No study could be found showing the contrary. These findings were also supported by a large multi-institutional MRI study in the UK (Lophatananon et al.: Re-evaluating the diagnostic efficacy of PSA as a referral test … in contemporary MRI-based image-guided biopsy pathways. Journal of Clinical Urology, published 12-01-2021). This clinical phenomenon of ‘prostate size matters’ should be mentioned in the manuscript because future studies may consider prostate volume as a parameter in their algorithms.

Response: Thank you for drawing our attention to this very important clinical phenomenon. Indeed, this is important information for future readers. A relevant paragraph has been added to the manuscript in Chapter 4.1.

Round 2

Reviewer 1 Report

The authors answered all comments and suggestions.

Reviewer 2 Report

The revised manuscript has adequately addressed the reviewer's comments and critiques. I support the publication of this revised manuscript.